# Effect of *Bacillus subtilis* Zeolite Used for Sediment Remediation on Sulfide, Phosphate, and Nitrogen Control in a Microcosm

**DOI:** 10.3390/ijerph19074163

**Published:** 2022-03-31

**Authors:** Maheshkumar Prakash Patil, Ilwon Jeong, Hee-Eun Woo, Seok-Jin Oh, Hyung Chul Kim, Kyeongmin Kim, Shinya Nakashita, Kyunghoi Kim

**Affiliations:** 1Industry-University Cooperation Foundation, Pukyong National University, 45 Yongso-ro, Nam-gu, Busan 48513, Korea; mahesh@pukyong.ac.kr; 2Department of Ocean Engineering, Pukyong National University, 45 Yongso-ro, Nam-gu, Busan 48513, Korea; jeongiw@pukyong.ac.kr (I.J.); hew119@pukyong.ac.kr (H.-E.W.); 3Department of Oceanography, Pukyong National University, 45 Yongso-ro, Nam-gu, Busan 48513, Korea; sjoh1972@pknu.ac.kr; 4Fisheries Resources and Environment Research Division, West Sea Fisheries Research Institute, National Institute of Fisheries Science, 14 Seonnyeobawi-ro, Jung-gu, Incheon 22383, Korea; hckim072@korea.kr; 5Coastal and Estuarine Sediment Dynamics Group, Port and Airport Research Institute, 3-1-1 Nagase, Yokosuka 239-0826, Kanagawa, Japan; seniorz@hiroshima-u.ac.jp; 6Graduate School of Advanced Science and Engineering, Hiroshima University, Kagamiyama 1-4-1, Higashi-Hiroshima 739-8527, Hiroshima, Japan; nakashita@hiroshima-u.ac.jp

**Keywords:** *Bacillus subtilis zeolite*, sediment remediation, acid volatile sulfite, marine sediment, microcosm experiment, ammonia

## Abstract

Eutrophication is an emerging worldwide issue concerning the excessive accumulation of various pollutants in sediments, owing to the release of industrial or household wastewaters to coastal areas. The coastal sediment of Goseong Bay in the Republic of Korea is organically enriched with pollutants, including heavy metals, sulfide, phosphate, and ammonia. Microbial remediation and capping techniques have been suggested as effective routes for sediment remediation. In this study, *Bacillus subtilis zeolite* (BZ) was used as a sediment capping material, and effective remediation of coastal sediment was observed in a 40-day laboratory microcosm experiment. A significant decrease in the sediment water content and reduced concentration of acid volatile sulfide were observed in the BZ-capped sediment. In the overlying water and pore water, significant decreases in phosphate and dissolved inorganic nitrogen (DIN; NO_2_-N + NO_3_-N and NH_4_-N) concentrations were observed in the BZ-treated experiment. Based on our findings, we conclude that BZ could be an effective capping material for coastal sediment remediation.

## 1. Introduction

Eutrophication due to excess accumulation or supply of organic matter in sediments is a worldwide problem. Nitrogen and phosphate are basic nutrients that control the growth of algae [1]. Dissolved inorganic nitrogen (DIN), such as nitrate, nitrite, and ammonia, in the aquatic system supports the synthesis of humic substances, amino acids, and urea by different reactions, which support the growth of phytoplankton and bacteria [2,3]. In sediment, sulfur is a major factor in the biogeochemical cycle, and its impact is related to carbon and oxygen cycle regulation. Sulfur deposits in sediment release hydrogen sulfite (H_2_S) upon acid treatment, or acidification processes [4]. H_2_S is naturally formed by decomposing organic matter in coastal sediments. Bacterial sulfate reduction also plays an important role in the mineralization of organic carbon in marine sediments [5]. Under oxygen-depleted sediment, sulfide cannot be oxidized, and combines with hydrogen to form H_2_S gas. H_2_S is toxic to living organisms and consumes oxygen when it is oxidized [4]. Acid volatile sulfide (AVS) is observed in the sulfide-rich zone of fine-grained anoxic sediments and is toxic to benthic organisms [4,6]. The environmental criterion for AVS in fish farm sediment is 2.5 mg/g dry sediment [7].

Phosphorous and nitrogen are vital components (growth factors) of the biosphere. However, owing to excessive availability, they cause eutrophication [8], which can result in decreased concentration of dissolved oxygen, and reduced oxidation-reduction potential. Significant increases in these pollutants in marine sediments cause harm to food resources, habitats and water quality. Owing to depletion of dissolved oxygen, benthic macro-organisms and micro-organisms cannot survive.

The capping of polluted marine sediments for remediation is well known technology [9,10,11]. In physicochemical-based capping techniques, sediments are covered (capped) by a layer of clean sediment, steel slag, and zeolites for contaminated sediment restoration [12]. Bioactive sediment capping materials have also been developed, in which the micro-organisms are capable of transforming sediment pollutants within caps [13].

Bacteria from the genus *Bacillus* are rod-shaped, facultative anaerobes that can grow in environments with high salt concentrations, and are known to produce numerous enzymes and acids and to conduct aerobic fermentation. *Bacillus subtilis* can form endospores and produce nitrate reductase. It is usually found in soil, but has also been reported to occur in the marine environment [14,15]. It has been reported that *B. subtilis* has a high affinity, and adhesion capacity, for minerals/zeolite [16], and is associated with removal of heavy metals by biosorption [17]. In addition, *Bacillus* spp. have been reported to solubilize phosphate in sediment [18] enzymatically (alkaline phosphatase). or by acid hydrolysis [19]. *Bacillus* spp. can increase nitrate reduction in the presence of iron oxides [20]. likely via their nitrate reductase enzymes [21]. Considering these significant characteristics of *B. subtilis* and the use of zeolite in the field of sediment management, we proposed the use of *Bacillus subtilis* zeolite (BZ) for sediment remediation.

In this study, we used BZ as a capping material to evaluate changes in pH, water content percentage, and concentrations of AVS, phosphate, and DIN in polluted sediment in a microcosm experiment.

## 2. Materials and Methods

### 2.1. Materials

Seawater and sediment were collected from Goseong Bay and immediately transported to the laboratory to perform a microcosm study on the same day. Goseong Bay is semi-enclosed and located in the Southeast Sea of the Republic of Korea. It is famous for shellfish farming, and its coastal sediment is highly enriched with organic matter and polluted with sulfides and heavy metals [22,23]. Before use in the experiment, seawater was treated with nitrogen gas to reduce the dissolved oxygen concentration to 2 mg/L or less. The BZ utilized in this investigation was bought from Sangsung Bio-tech in Ulsan, South Korea, under the brand name N-POWER-1. BZ with ca. 2 mm in diameter after adhesion of *B. subtilis* with a cell density of 3 × 10^4^ colony-forming units per gram of zeolite.

### 2.2. Microcosm Experiment

The experiment was designed using a 3 L plastic PET jar with a lid. Two sets were prepared in duplicate, with the first set as a control and the second set for BZ capping on sediment. In the first set (control), 1500 mL of sediment was added, the jar was filled with seawater up to the neck, and then the jar was closed with a lid, and sealed with parafilm tape to avoid air exchange. In the second set (BZ), after adding 1350 mL of sediment to each jar, 150 mL of BZ was added, followed by the addition of seawater up to the neck of the jar, and then the lid was closed and sealed with parafilm (Sigma Aldrich, Burlington, MA, USA). All jars were kept in a water bath at 20 °C, and a microcosm study was performed for 40 days. Samples from each set were collected on days 0 (initial), 1, 3, 5, 10, 20, and 40 for analysis. Overlying water (OW) samples were collected using a syringe and pore water (PW) samples were prepared by centrifuging the sediment at 3400 rpm for 15 min at 20 °C. Before analysis, both OW and PW filtrate was collected with a 0.45 µm syringe filter. All samples for analysis were kept in an airtight container and at 4 °C in the dark until analysis.

### 2.3. Experimental Procedures

A pH meter (Horiba D-50) was used to measure pH. The AVS content of the sediment sample was analyzed on the same day, using an Hedrotech-S Kit [24]. The NH_4_-N, NO_2_-N + NO_3_-N, and PO_4_-P concentrations were determined according to standard methods [25], using an autoanalyzer (SWATT, BLTEC). The sediment sample was dried for 6 h at 100 °C, and the water content was calculated using the following equation (Equation (1)):(1)$$\mathrm{Water}\;\mathrm{content}\left(\%\right)\;=\;\frac{\mathrm{Wet}\;\mathrm{Sample}\;\mathrm{Weight}\;-\times \;\mathrm{Dry}\;\mathrm{Sample}\;\mathrm{Weight}\;}{\mathrm{Dry}\;\mathrm{Sample}\;\mathrm{Weight}}\times \;100$$

## 3. Results

The pH of the OW of the control and experiment (BZ) decreased from alkaline to acidic, as shown in Figure 1a. The pH ranges in the control and BZ-capped sediment were 6.87–8.10 and 6.79–8.10, respectively. In the PW, the pH slightly decreased and then increased up to 20 days in the control and up to 5 days in the BZ (Figure 1b), and a rapid decrease in the pH in the control and in the experiment were observed on day 40. There was no significant variation in pH in the PW. A decrease in water content was observed after 1 day in the BZ sediment compared with the initial results (Figure 1c). The water content in the sediment capped with BZ indicated a sharp decrease on day 3 (ca. 145%) until the end of the experiment, except for on day 20. At the end of the experiment (after 40 days), the BZ-capped sediment showed a sharp decrease in water content of up to 148%, whereas that in the control was 204%.

A considerable reduction in AVS was observed in the BZ-capped sediment compared with that in the control sediment. As shown in Figure 1d, the AVS concentration in dry sediment ranged from 0.014 mg S/g to 0.226 mg S/g in the control and from 0.008 mg S/g to 0.068 mg S/g in BZ-capped sediments.

The changes in the phosphate concentrations in OW and PW are presented in Figure 2. In the OW, the phosphate concentration was higher in the BZ-capped experiment than in the control group throughout the experimental period. However, in the PW, the phosphate concentration of BZ-capped sediment decreased up to day 20 and increased on day 40, compared with that in the control sediment samples.

The NO_2_-N + NO_3_-N concentration in the OW (Figure 3a) and PW (Figure 3b) of the control and BZ-capped experiments were evaluated. The NO_2_-N + NO_3_-N concentration in the OW ranged from 0.05 mg/L to 3.35 mg/L in the control, and from 0.05 mg/L to 2.13 mg/L in the BZ-capped sediment. Compared with the control, BZ-capped OW showed a significant decrease in NO_2_-N + NO_3_-N concentration up to day 40. However, a significant difference in the NO_2_-N + NO_3_-N concentration was not observed in the PW between the control and BZ-capped sediments.

A decrease in NH_4_-N concentration in the OW and PW from the BZ-capped experiment were observed, compared with that in the control experiment (Figure 3c,d). The NH_4_-N concentration ranged from 0.36 mg/L to 4.35 mg/L in the OW and from 0.49 mg/L to 6.29 mg/L in the PW of the control. Meanwhile, that in the BZ-capped sediment ranged from 0.36 mg/L to 2.64 mg/L in the OW and from 0.49 mg/L to 6.73 mg/L in the PW. In the OW, a decrease in NH_4_-N concentration from days 10 to 40 was observed in the BZ-capped experiment. Similarly, in the PW, NH_4_-N concentration decreased up to day 20 and then increased on day 40, compared with that of the control.

## 4. Discussion

The average pH of oceans is approximately 8.2. Continuous absorption of CO_2_ causes pH to decrease and the ocean to become acidic. Sediment pH is generally low, because the decomposition of nutrient-enriched matter produces organic acids [26]. In this study (Figure 1), the pH of the OW decreased, because there was no gaseous exchange in the microcosm experiment. *B. subtilis* is well known for producing many acids, and the acid produced by *B. subtilis* has been reported to enhance the flocculation of organic and inorganic suspensions in the presence of various cations. Moreover, decreases in pH due to flocculation have been reported [27]. As shown in Figure 1c, the water content decreased after capping. Thus, the capping layer of BZ on the sediment may have a consolidation effect and decrease the water content by releasing water from the sediment. Consolidation of BZ-capped sediment also leads to suppression of sediment resuspension by physical effects, such as currents and waves [28].

A reduction in AVS was observed in the BZ-capped sediment (Figure 1d). *Bacillus* species have been reported to remove sulfate facultatively and strictly anaerobically from acid mine wastewater [29]. *Bacillus subtilis* can grow anaerobically, either by fermentation or by using nitrite or nitrate as a terminal electron acceptor [30]. These studies support the hypothesis that *B. subtilis* can grow well in oxygen-depleted layers of sediment and reduce AVS.

Phosphate concentration in the PW decreased in the BZ-capped sediment (Figure 2). It is well known that several bacteria play important roles in phosphate solubilization enzymatically, or by acid hydrolysis; *Bacillus* is one of them [18,19]. The sharp increase in the release of PO4-P in BZ-capped sediment on day 40 was due to a decrease in the pH of the sediment, which may enhance nutrient availability and the formation of organic acids. It is generally accepted that phosphate solubilization by bacteria is associated with the release of organic acids [31].

A significant decrease in the DIN concentration in the OW and PW in the BZ-capped sediment was observed (Figure 3). Micro-organisms play an important role in nitrogen cycling in the oxygen-depleted zone, whereas nitrate reduction to nitrite and dissimilatory nitrate reduction to ammonia are major mineralization pathways [32]. *B. subtilis* grows aerobically and anaerobically by fermentation, or by using nitrite or nitrate as a terminal electron acceptor [30]. In the experimental sites, aerobic and anaerobic cyclic repetition of denitrification processes may enhance the decrease in DIN [33]. Goseong Bay sediment is highly contaminated with metals (including iron) [22,23], and *Bacillus* species have been reported to enhance nitrate reduction in the presence of iron [20], which supports our finding of decreased DIN in BZ-capped sediments in laboratory microcosm experiments.

## 5. Conclusions

This study investigated the efficacy of BZ as a capping material to control sediment pollutants in a laboratory microcosm experiment. Decrease in pH in the BZ treated OW was observed, while there was no significant variation in pH in the PW. In addition, water content decreased, due to the consolidation effect and the release of water from the sediment. The results indicate that BZ significantly decreased AVS, phosphate, and DIN in Goseong Bay sediment. Based on these findings, we propose that BZ could be an effective capping material for marine sediment remediation. Further studies on the optimization of BZ application for sediment remediation are required.

## Figures and Tables

**Figure 1 ijerph-19-04163-f001:**
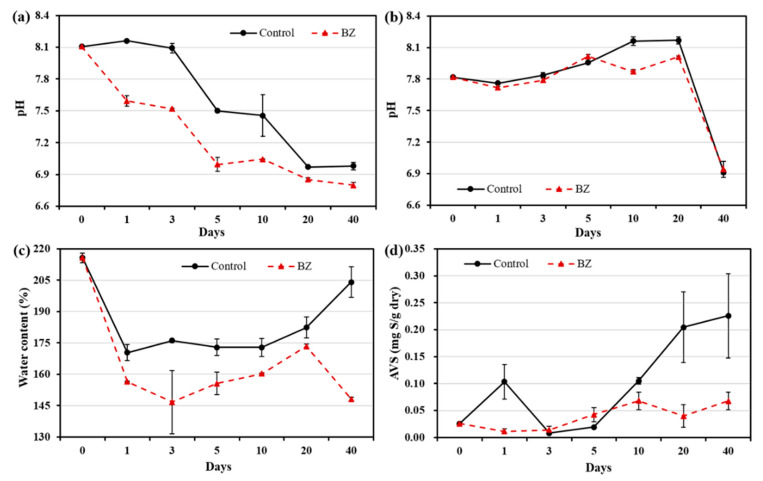
Changes in pH at different times in (**a**) overlying water, (**b**) pore water, and temporal changes in (**c**) water contents and (**d**) AVS. All analytical measurements were performed in triplicate and error bars indicate the standard deviation of the mean.

**Figure 2 ijerph-19-04163-f002:**
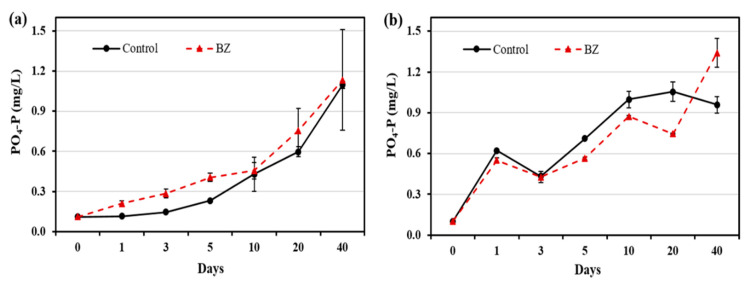
Changes in phosphate concentration at different times in (**a**) overlying water and (**b**) pore water. All analytical measurements were performed in triplicate and error bars indicate the standard deviation of the mean.

**Figure 3 ijerph-19-04163-f003:**
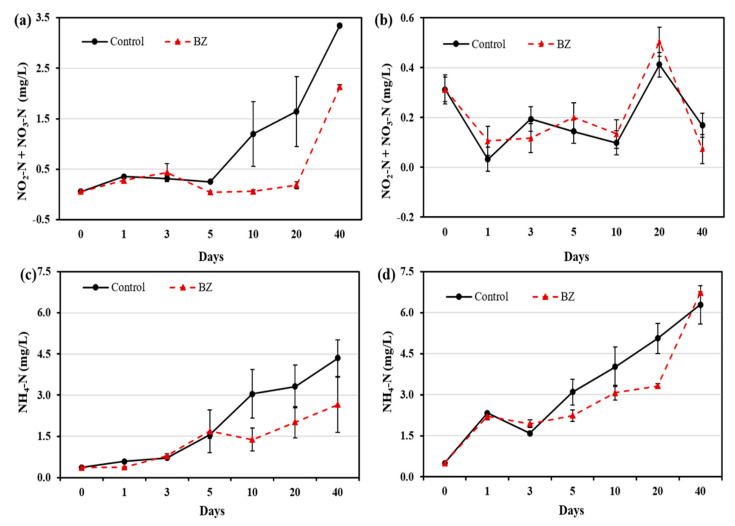
Changes in nitrogen-containing compound concentration at different times in (**a**) overlying water (**b**) pore water, and changes in ammonium concentration at different times in (**c**) overlying water and (**d**) pore water. All analytical measurements were performed in triplicate and error bars indicate the standard deviation of the mean.

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
