# Peer review of "Effect of Bacillus subtilis Zeolite Used for Sediment Remediation on Sulfide, Phosphate, and Nitrogen Control in a Microcosm"

_ijerph, 2022, doi:10.3390/ijerph19074163_

Round 1
Reviewer 1 Report
The manuscript presents a 40 days trial with Bacillus subtilis zeolite for sediment remediation, well described the results and interesting. I recommend accepting this manuscript after minor revision. Needed to address the following point:
1. Author used only a single concentration of BZ in their 40 days experiment, if possible provide the effect of different concentrations of BZ on sediment remediation.
2. Is there any other microbial species/strains tested for comparative analysis on sediment remediation?
Reviewer 2 Report
The experimental setup seems adequate and the results give sufficient evidence to suggest that BZ can be used for coastal sediment remediation - theoretically.
However, I cannot se how treatment with B. subtilis can be administered on an ecosystem in a way that is useful.
The Discussion leaves this important aspect to the reader´s imagination as now clue is provided. I am quite critical to this and doubt the practical value of the study for this reason.
Reviewer 3 Report
My most serious concern deals with the confirmation of bioremediation process with adding the bacterial strains. Novel research needs checking many factors to confirm the biodegradation studies.
I recommend the authors to study the quality and quantity of enzyme activity in the treatment and control/ Furthermore, the checking of CO2 production via sturm test is will be good to present the activity of this microbe in the bioremediation process.
Other minor revision are included as:
- The study was done in vitro so please don’t use in situ sentence
- Line 173, Spill out the genus name in the beginning of sentence
- Try to use the new references in the field. There are so many unneeded references in the text.
- There is mostly no discussion, try to discuss what new you can add to previous results and what in detail has been observed previously
Reviewer 4 Report
- It is suggested to add important experimental data in the abstract.
- It is suggested to further emphasize the innovation of the work in the introduction.
- The title of section 2.3 is suggested to be changed to ‘Experimental procedures’.
Round 2
Reviewer 2 Report
The manuscript is now well written, and has a sound experimental design. Thus, the manuscript provides useful information on the use of Bacillus subtilise Zeolite for Sediment Remediation. This can be practically used in a microcosm system. I doubt that the practical value goes beyond closed or semi-enclosed systems, yet the value is still significant.
Reviewer 3 Report
Authors had addressed the main comments and i recommend to accept in present form
This manuscript is a resubmission of an earlier submission. The following is a list of the peer review reports and author responses from that submission.
Round 1
Reviewer 1 Report
- "2.2. Microcosm Experiment" The authors should mention how many samples were in each set.
- "2.2. Microcosm Experiment" I feel that there are too few controls in this experiment. Why did the authors only use two sets (sediment with and without BZ)? Where is the negative control (water without sediment - maybe the glass bottle adheres particles that contain phosphates and the concentration in the water goes down)? And so on (maybe water with only BZ as positive control). Maybe the experiment needs to be redone with more controls.
- "4.Discussion" I feel that there should have been different sets with different amount of BZ to actually show what quantity could support the authors claims that "Thus, the capping layer of BZ on the sediment may have a consolidation effect" - Maybe the authors are using a very high amount of BZ and this could work with less. It is important for future remediation strategies to know how much BZ to use.
- "Based on these findings, we propose that BZ is an effective capping material for marine sediment remediation." I cannot accept this conclusion (even if it is based on what the authors have presented) because what the authors have presented are incomplete data. There needs to be some additional controls in order to have this type of conclusion. I suggest that the authors redo the experiment with a negative and positive control and see what type of sets they will use for the different quantity of BZ in order to be able to draw this type of conclusion.
- The idea of the article is great. It just needs a bit more work.
Reviewer 2 Report
This paper is useful to remove eutrophication species of S,P and N using B. subtilis zeolite for the consideration of sustainability.
However, please check the following points.
Line 28,99,115,117
d →day or days
Abbreviation
Please write the first appearance part.
Line 102 to 101 pore water (PW)
In Materials section
Please write the analysis of B. subtilis in material section clearly.
Also please write what kind of zeolite was used and show the reason of this zeolite utilization.
Fig. 3 (a), (b)
In Y axis please use NO2-N + NO3-N.
In Discussion parts
Please add the mechanism of remediation.
In conclusion
Please write in the point and point more clearly.
Round 2
Reviewer 1 Report
You have resolved the conclusions and the overall article looks better now.
Unfortunately I feel that there is too few replicates per set and not enough controls used to merit the publication of this research in it's current state.
Please take care and add more to this article (different quantities of zeolite, different concentrations of bacteria inoculated and so on) so that this article could be elevated more.